# Theory-agnostic searches for non-gravitational modes in black hole ringdown

Francesco Crescimbeni,[1, 2, *] Xisco Jimenez Forteza,[3, †] Swetha
Bhagwat,[4, ‡] Julian Westerweck,[4, 5, 6, §] and Paolo Pani[1, 2, ¶]

[1]*Dipartimento di Fisica, Sapienza Università di Roma, Piazzale Aldo Moro 5, 00185, Roma, Italy*
[2]*INFN, Sezione di Roma, Piazzale Aldo Moro 2, 00185, Roma, Italy*
[3]*Departament de Física, Universitat de les Illes Balears, IAC3 – IEEC, Crta. Valldemossa km 7.5, E-07122 Palma, Spain*
[4]*School of Physics and Astronomy & Institute for Gravitational Wave Astronomy,*
*University of Birmingham, Birmingham, B15 2TT, United Kingdom*
[5]*Max-Planck-Institut für Gravitationsphysik (Albert-Einstein-Institut), Callinstraße 38, D-30167 Hannover, Germany*
[6]*Leibniz Universität Hannover, D-30167 Hannover, Germany*

In any extension of General Relativity (GR), extra fundamental degrees of freedom couple to gravity. Besides deforming GR forecasts in a theory-dependent way, this coupling generically introduces extra modes in the gravitational-wave signal. We propose a novel theory-agnostic test of gravity to search for these nongravitational modes in black hole merger ringdown signals. To leading order in the GR deviations, their frequencies and damping times match those of a test scalar or vector field in a Kerr background, with only amplitudes and phases as free parameters. By applying this test to GW150914, GW190521, and GW200129, we find no strong evidence for an extra mode; however, its inclusion modifies the inferred distribution of the remnant spin. This test will be applicable for future detectors, which will achieve signal-to-noise ratios higher than 100 (and as high as 1000 for space-based detectors such as LISA). Such sensitivity will allow measurement of these modes with amplitude ratios as low as 0.02 for ground-based detectors (and as low as 0.003 for LISA), relative to the fundamental mode, enabling stringent agnostic constraints or detection of scalar/vector modes.

## I. INTRODUCTION

The black hole (BH) spectroscopy program [1–4] plays a prominent role in the landscape of strong-field tests of General Relativity (GR) [5–8] and provides a unique method for examining the nature of compact remnants formed post-coalescence [9]. This program focuses on extracting the remnant quasinormal modes (QNMs) [10–13] during the ringdown phase of a binary merger. In the context of linear perturbation theory, the gravitational-wave (GW) signal $h(t)$ at intermediate times after the merger is represented by a superposition of the QNMs of the remnant [14], as it transitions towards a stationary configuration. Schematically,

$$h(t) = \sum_i A_i \cos\left(2\pi f_i t + \phi_i\right) e^{-\frac{t}{\tau_i}}, \qquad (1)$$

where $A_i$, $\phi_i$, $f_i$, $\tau_i$ are the amplitude, phase, frequency, and damping time of the $i$-th QNM, whereas $i \equiv (l, m, n)$ collectively represents the multipolar, azimuthal, and overtone index, respectively.

If the remnant is a BH, GR predicts that the infinite spectrum of QNMs is uniquely determined by its mass and spin $(M_f, \chi_f)$. This provides opportunities for conducting multiple null-hypothesis tests of gravity [15, 16] and investigating the nature of the remnant [17–19].

As suggested by Lovelock's theorem [20, 21], an almost unavoidable ingredient of theories beyond GR is the presence of extra degrees of freedom nonminimally coupled to gravity [6, 22]. Examples are ubiquitous and include scalar fields in scalar-tensor theories and Horndeski's gravity [23] (and their vector counterpart [24]), high-curvature corrections to GR that predict extra (pseudo)scalars and dilaton fields [25–27], Einstein-Aether [28] and Hořava–Lifshitz [29] gravity that postulate an extra timelike vector field, and massive gravity [30, 31] with both scalar and vector dynamical degrees of freedom (see [6] for a review of GR extensions and their field content). Effective extra degrees of freedom are also unavoidable in any approach that treats GR as the leading order term in an effective-field-theory expansion (e.g., [27]) and in low-energy effective string theories. These nonminimally coupled fields may modify the stationary BH solutions, leading to deviations from the Kerr metric, and/or modify the dynamics of the theory. In either case, two generic predictions are: i) a deformation of the Kerr QNMs,

$$f_i = f_i^{\text{Kerr}}(1 + \delta f_i), \qquad \tau_i = \tau_i^{\text{Kerr}}(1 + \delta \tau_i), \quad (2)$$

and ii) the existence of extra modes in the gravitational signal, that can be excited during the ringdown. This second option is due to the fact that the nonminimal coupling between new degrees of freedom and gravity results in coupled systems of linear perturbation equations, which act as a coupled set of oscillators [32–35]. It is therefore natural to split the ringdown signal (1) into

---

[*] francesco.crescimbeni@uniroma1.it
[†] f.jimenez@uib.es
[‡] s.bhagwat@bham.ac.uk
[§] j.m.westerweck@bham.ac.uk
[¶] paolo.pani@uniroma1.it

two contributions,

$$h(t) = \sum_i A_i \cos\left(2\pi f_i^{\text{Kerr}}(1+\delta f_i)t + \phi_i\right) e^{-\frac{t}{\tau_i^{\text{Kerr}}(1+\delta\tau_i)}}$$
$$+ \sum_i \hat{A}_i \cos\left(2\pi \hat{f}_i t + \hat{\phi}_i\right) e^{-t/\hat{\tau}_i}, \qquad (3)$$

where we use the hat to denote quantities related to the extra modes.

Standard tests of gravity based on the ringdown are rooted in the first line of Eq. (3). Namely, they are aimed at measuring $\delta f_i$ and $\delta\tau_i$ and check whether they are compatible with the null hypothesis [5]. This can be accomplished in two complementary ways, either through a theory-agnostic or theory-dependent method. However, both approaches have their own limitations. In the theory-agnostic approach one uses the parametrization (2), where $f_i^{\text{Kerr}}$ and $\tau_i^{\text{Kerr}}$ are known functions of the BH mass and spin, while $\delta f_i$ and $\delta\tau_i$ depend on the mass, spin, and all extra fundamental coupling constants of the theory. We will focus on small deviations from GR. This is motivated by an effective-field-theory perspective wherein Einstein's theory is the lowest term in an expansion containing all possible higher-order operators, and also by the stringent observational constraints already placed by GW observations. As a result, any deviation from the standard GR prediction must be small and proportional to (powers of) the coupling constants of the theory. In particular, $\delta f_i, \delta\tau_i \ll 1$ and are proportional to some combination of the mass and coupling constants. However, even in this case they are still generic, theory-dependent functions of the spin $\chi_f$. Current parametrizations either neglect such spin dependence [5, 36, 37] or consider a small spin expansion of each deviation [38–40], which inevitably inflates the number of free parameters in the model. In the theory-dependent approach, the QNMs are computed in a given theory of gravity. For most theories this can be done again only in the small-coupling limit and very often perturbatively in the spin [32, 41–50]. To reach good convergence, one needs to push the spin expansion to very high order [49], which is very challenging from the technical point of view. Alternatively, one needs to solve intricate systems of coupled partial differential equations [51–56]. This approach has the benefit of limiting the number of free parameters to the sole coupling constants and BH spin, but must be performed on a case-by-case basis for every given theory.

Given the above limitations, it would be highly desirable to develop complementary ringdown tests which are both theory-agnostic and accurate. In this work we explore a currently unbeaten path, related to the second line of Eq. (3). Namely, we propose to look for extra modes in the ringdown signal. Note that these extra modes are unavoidably present in beyond-GR theories, raising the important issue that current ringdown analyses (based only on the first line of Eq. (3)) are incomplete.

For concreteness, let us consider the case of an extra scalar degree of freedom nonminimally coupled to gravity (the same argument applies to other types of fields).

Due to the coupling, the gravitational perturbations will contain also scalar modes (e.g., [32, 34, 42, 57] for two concrete examples in theories with quadratic curvature terms),

$$\hat{f}_i = f_i^{\text{Kerr}, s=0}(1 + \delta\hat{f}_i), \qquad \hat{\tau}_i = \tau_i^{\text{Kerr}, s=0}(1 + \delta\hat{\tau}_i),$$
$$(4)$$

where $f_i^{\text{Kerr}, s=0}$ and $\tau_i^{\text{Kerr}, s=0}$ are the QNMs of a test scalar field in the Kerr metric, and also in this case $\delta\hat{f}_i$ and $\delta\hat{\tau}_i$ are complicated, theory-dependent, functions of the mass, spin, and coupling constants, which incorporate both deviations from the GR BH background and modified dynamics. Crucially, in this case the amplitudes $\hat{A}_i$ of these modes are *proportional* to (powers of) the coupling constants [32, 34, 35, 42, 57], and they must vanish in the GR limit. Therefore, to leading order in the corrections, we can neglect $\delta\hat{f}_i$ and $\delta\hat{\tau}_i$, so that the GR deviations are generically parametrized only by the amplitude of the *test-field* modes on a GR BH background. This is precisely what happens in so-called dynamical Chern-Simons gravity [26, 32, 34] (see also Appendix VII A), although it is a generic feature [35].

The above considerations suggest a novel ringdown test of gravity based on the following waveform model

$$h(t) = \sum_i A_i \cos\left(2\pi f_i^{\text{Kerr}}t + \phi_i\right) e^{-\frac{t}{\tau_i^{\text{Kerr}}}}$$
$$+ \sum_i \hat{A}_i \cos\left(2\pi f_i^{\text{Kerr}, s=0}t + \hat{\phi}_i\right) e^{-\frac{t}{\tau_i^{\text{Kerr}, s=0}}} \quad (5)$$

where for simplicity we have neglected the GR deviations in the first line, since those are very well studied by standard ringdown tests (and subjected to the aforementioned limitations). Notably, as explained below and although our framework is generic, one of its advantages is that it can be applied to a single multipole couple $(l, m)$, in which case $i = 0, 1, 2, ..$ would simply correspond to the overtone number $n = 0, 1, 2, ..$ at fixed $(l, m)$. If one includes only the dominant GR fundamental mode in the analysis ($i \equiv (2, 2, 0)$ in the first line of Eq. (5)), as we will do below for the so-called GR0+S or GR0+V models, the corrections $\delta f_{220}$ and $\delta\tau_{220}$ to the fundamental gravitational QNM are degenerate with the final mass and spin. Therefore, they can be neglected without loss of generality. Instead, including also gravitational overtones would require adding extra corrections $\delta f_i$ and $\delta\tau_i$ (with $i \equiv (2, 2, n \geq 1)$) in the first line of Eq. (5), as typically done in ordinary BH spectroscopy tests with overtones. In the models considered here, the mentioned degeneracy is broken as we neglect higher harmonics but include at least two modes through either an overtone (GR1 model) or an extra scalar/vector mode (GR0+S/GR0+V) in addition to the fundamental GR mode, determining the final mass and spin uniquely. Explicitly, our models contain the modes: GR1 → $(2, 2, 0), (2, 2, 1)$; GR0+S → $(2, 2, 0), (2, 2, 0)_{\text{scalar}}$; GR0+V → $(2, 2, 0), (2, 2, 0)_{\text{vector}}$.

In practice, here we will focus on a standard GR ringdown waveform (first line of Eq. (5)) augmented by new extra modes. Remarkably, to leading order these extra modes are *known* functions of the BH mass and spin, since they are those of a free test (scalar, vector, etc) field propagating on the Kerr metric (see, e.g., [58, 59] for tabulated values). This allows searching for extra modes in a theory-agnostic way, where the amplitudes and phases of the extra modes are the only beyond-GR parameters. In this sense, this test is reminiscent of searches for extra (scalar, vector) polarizations in GW signals in a theory-agnostic fashion [5, 60–62] and is complementary to ordinary ringdown tests (see, e.g., [5, 36, 37, 63–66]), or to test with multiple free modes [5].

## II. SEARCHING FOR EXTRA RINGDOWN MODES

The ringdown signal comprises of two polarizations and the modes are decomposed in a basis of spin-weighted spheroidal harmonics that depend on the remnant spin inclination angle [66]. In a non-precession quasicircular coalescence, $(lmn) = (220)$ is the dominant mode. For concreteness, here we focus on the most interesting quadrupolar ($l = 2$) case and neglect spin precession of the progenitor binary, but our test can be applied also to higher-order modes and to the precessing case (see Appendix VII B). In particular, our ringdown waveform model has the following parameters:

$$\underline{\theta} = \{M_f, \chi_f, A_{22j}, \phi_{22j}, \hat{A}_{220}^{s=0,1}, \hat{\phi}_{220}^{s=0,1}\} \qquad (6)$$

where $j = 0, \ldots, N$, with $N$ the total number of overtones, whereas $\hat{A}_{220}^{s=0,1}$ and $\hat{\phi}_{220}^{s=0,1}$ are the amplitude and phase of the extra scalar ($s = 0$) or vector ($s = 1$) (220) mode. We will dub a model with $N$ tones and no extra mode GRN, whereas we will denote as GRN+S (GRN+V) a model with $N$ gravitational tones and an extra scalar (vector) mode. In practice, we define the amplitude ratio $A_R$ of a given mode relative to the gravitational fundamental one. Notice that in general, these amplitudes depend on the specific theory (see an example for dynamical Chern-Simons theory in the Appendix VII A), and estimating their value in a merger requires performing nonlinear coalescence simulations beyond GR. For this reason, it is not easy to predict the relevant importance of extra modes with respect to other GR effects, e.g., overtones or nonlinearities. For the sake of generality, here we remain agnostic on the amplitude modeling. We fix the luminosity distance $d_L$, the sky location of the system, and the inclination angle as done standard ringdown analysis [5, 15, 39, 67]. Although marginalization would be the ideal approach, it involves its own challenges, like the accurate excision of the pre-merger part to avoid the contamination of the post-merger data. Failing on the accurate excision could impact negatively on the accuracy of the no-hair theorem tests performed using the post-merger data [68, 69]. Furthermore, the sky-

marginalization has been shown to introduce only mild differences for current GW events, from [69, 70] where the results obtained, are still consistent with those of [15, 39] which fixed the sky location to the maximum likelihood values. However, the scenario might be different for higher-SNR events and with the inclusion of higher harmonics. Note that, for small-redshift sources (including those detected so far and presumably the loudest events detected in the future) $d_L$ is degenerate with the overall ringdown amplitude and can therefore be neglected without loss of generality. However, the possibility of neglecting inclination angle is a prerogative of our test, since it can involve only ($l = m = 2$) modes , which have the same pattern functions and spheroidal-harmonic decomposition (similarly to overtone-based tests but without the issues related to overtones). Compared to common tests based on subleading modes of different $(l, m)$, this allows for the practical simplification of sampling in only one of these three degenerate parameters. The test can be expanded to include higher harmonics, in which case one would have also to include the inclination as an extra parameter.

The frequencies and damping times of the relevant modes are shown in the Appendix VII C. At variance with overtones, the frequency of the (220) scalar or vector mode is always well separated[1] from that of the fundamental gravitational mode (and hence more easily resolvable from the latter), while the damping time is comparable (and hence the mode survives longer than overtones in the signal, almost as long as the fundamental gravitational mode).

## III. BAYESIAN ANALYSIS ON REAL DATA

We exemplify our test on real events by performing a Bayesian parameter estimation using the PyCBC Inference code infrastructure [71]. The analysis aims to compute the posterior distribution of the parameters (6). We apply this test to three events: (i) GW150914 [72], the first GW event ever detected by LIGO (and still so far the one with the largest ringdown signal-to-noise ratio (SNR)), for which some debated evidence of overtones has been reported [15, 36, 66, 70, 73–80]; (ii) GW190521 [81], a peculiar event in the upper mass gap for which a tentative detection of the (330) and other modes (and possibly precession) has been obtained [67, 82] and which is prone also to ringdown amplitude-phase consistency tests [65]; (iii) GW200129, a peculiar loud event showing some tension with GR in some inspiral-merger-ringdown tests [5], tentatively ascribed to mismodelling of precession [83, 84]. As a proof of principle for the test, our parametrization assumes

––––––––

[1] Except possibly in the $\chi_f \to 1$ limit, which is not relevant for the spin values considered here.

plane reflection symmetry, which is valid for spin-aligned progenitor binaries. In the Appendix VII B, we show that relaxing this assumption does not affect the results for GW190521 and GW200129. For the luminosity distance, inclination, and sky location we adopt the maximum likelihood values reported in [85].

We use a gated-and-inpainted Gaussian likelihood noise model [67, 86, 87] to remove the influence of the pre-peak/non-ringdown times. The strain data within a time interval $t \in [t_c + t_{\text{offset}} - 0.5s, t_c + t_{\text{offset}}]$ are replaced/inpainted such that the filtered inverse power spectral density is zero at all the times corresponding to the chosen interval [87]. Here, $t_c$ is the coalescence time, while $t_{\text{offset}}$ defines the time in which we start our ringdown analysis.

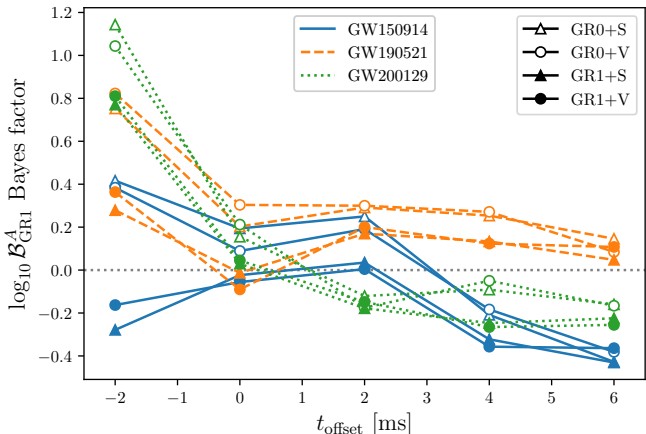

FIG. 1. $\log_{10}$ Bayes factors for various ringdown models with extra scalar or vector modes (labelled with 'A') with respect to the GR1 model as a function of the offset time $t_{\text{offset}}$. Different colors and line styles denote different events, while different markers show the chosen model. Note that each event has a different mass thus, a different time scale $t_{\text{offset}}$. Therefore, the $\log_{10}$ Bayes factor trends, specially at negative times, are expected to differ.

Our main results are summarized in Fig. 1, presenting the statistical evidence for different waveform models for these events and for different choices of $t_{\text{offset}}$. We show $\log_{10} \mathcal{B}^{\text{A}}_{\text{GR1}}$, where the Bayes factor $\mathcal{B}^{\text{A}}_{\text{GR1}}$ is the ratio between the evidence of a given model ('A') and that of GR1 (i.e., a model containing only the fundamental gravitational mode and the first overtone). According to Jeffreys' scale criterion [4, 76, 89, 90], a $\log_{10}$ Bayes factor larger than 1 (resp., 2) would imply a strong (resp., decisive) Bayesian evidence in favor of a given model relative to GR1. The small values of $\log_{10} \mathcal{B}^{\text{A}}_{\text{GR1}}$ shown in Fig. 1 for any $t_{\text{offset}}$ indicate that all models with extra scalar or vector modes have the same evidence as the GR1 one, presumably because the SNR in the ringdown for these events is not sufficiently high to exclude the presence of an extra scalar mode. This is consistent with what we shall discuss below with synthetic data.

Interestingly, despite Fig. 1 showing that there is no

| Model | $M_f(M_\odot)$ | $\chi_f$ | $A_{220} \times 10^{20}$ | $A_{R,220}$ |
|---|---|---|---|---|
| GR1 | $58^{+27}_{-22}$ | $0.35^{+0.49}_{-1.15}$ | $0.72^{+0.98}_{-0.50}$ | - |
| GR0+S | $52^{+35}_{-16}$ | $-0.18^{+0.94}_{-0.69}$ | $0.69^{+1.32}_{-0.57}$ | $0.80^{+2.74}_{-0.64}$ |
| GR1+S | $52^{+33}_{-16}$ | $-0.11^{+0.84}_{-0.76}$ | $0.72^{+1.30}_{-0.60}$ | $1.95^{+1.85}_{-1.75}$ |

TABLE I. 90% credible intervals for some of the parameters of event GW150914, assuming $t_{\text{offset}} = 2\,\text{ms}$ (see Appendix VII C for the posterior distributions).

statistical evidence for an extra scalar or vector mode, its inclusion affects the posterior distributions of the parameters. This is shown in Table I for a representative example of GW150914 analyzed with an extra scalar mode, where we denote the scalar-to-tensor mode amplitude ratio as $A_{R,220} = \hat{A}^{s=0}_{220}/A_{220}$. These results are also consistent with the full inspiral-merger-ringdown constraints provided in [5]. However, as expected, our peak values are shifted compared to those computed using a GR waveform approximant. Other events, different time offsets, or the vector case show qualitatively similar results. While the presence of an extra mode does not affect the distribution of the remnant mass significantly, it contributes to broadening that of the spin towards smaller values (see Appendix VII C for the posterior distributions). This generic feature can be understood from the fact that an extra (220) scalar/vector mode has a damping time comparable to the fundamental gravitational mode (and the damping time is only mildly sensitive to the spin for $\chi_f \lesssim 0.8$) and also has a higher frequency than the (220) and (221) gravitational modes. Thus, interpreting the overtone frequency with an extra scalar/vector mode requires a smaller remnant spin. Also, the amplitude of the fundamental gravitational mode is affected by the extra mode: in the GR0+S and GR1+S models the peak of the $A_{220}$ distribution is smaller because part of the information is contained in the scalar mode. Consequently, the amplitude ratio $A_{R,220}$ between the (220) scalar mode and the (220) gravitational mode peaks at some nonzero value. Finally, we do not find *strong* support for the presence of an additional mode for $t_{\text{offset}} > -2$ms. As in previous related work [77, 91], we observe that the Bayes factor tends to increase when the analysis is extended to negative times, since the model begins to fit higher overtones and nonlinear contributions. We interpret this increase not as evidence for extra modes, but rather as a consequence of model mismatch arising from applying the ringdown model *before* the merger, which can yield spurious results. Furthermore, for the event GW200129 – where we observe the largest values of $\log_{10} \mathcal{B}^A_{\text{GR1}}$ – data quality issues [92] and possible signatures of orbital eccentricity and/or precession [93, 94] are known to affect the inference of physical parameters. These factors could also contribute to the early-time rise of the Bayes factor in support for additional modes. A detailed investigation of these effects will be presented in future work.

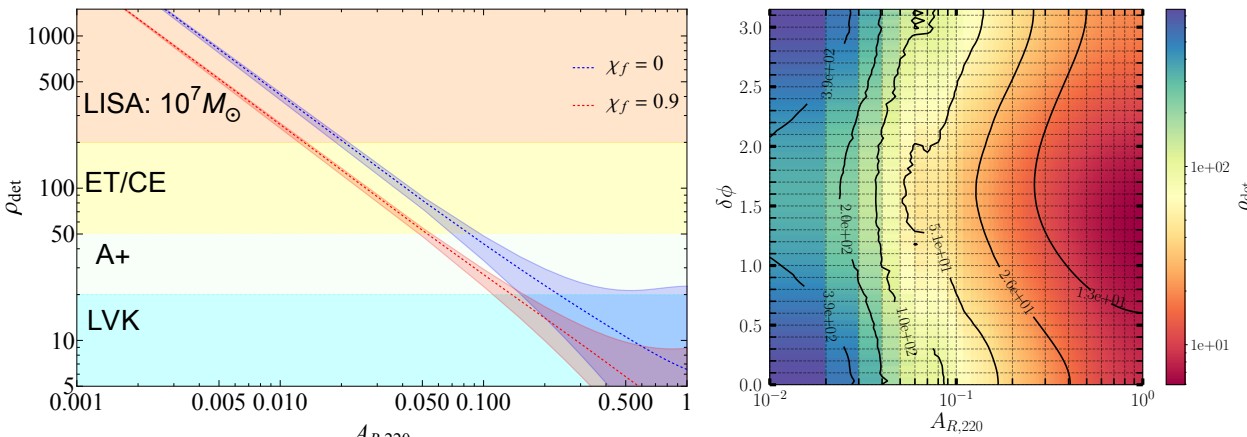

FIG. 2. Left panel: Minimum SNR necessary for detecting a scalar mode at $1-\sigma$ confidence level, according to the detectability criterion outlined in [3, 88] using a Fisher matrix approximation, which makes our estimates approximately independent of the sensitivity-curve profile. The blue and red dashed curves denote a different remnant spin $\chi_f$ with the phase difference between the GR and scalar modes fixed to $\delta\phi = 0$. The dependence on $\delta\phi \in [0, 2\pi]$ is bracketed by the corresponding colored bands. The horizontal shaded bands represent the expected ringdown SNR for a GW150914-like event with a remnant mass of $M_f = 62 \, M_\odot$ as observed by ground-based detectors, including the LIGO-Virgo-KAGRA network (light blue), A+ (light green), and CE/ET (yellow). The orange band corresponds to the expected ringdown SNR for a similar event with a remnant mass of $M_f = 10^7 \, M_\odot$ observed by LISA (orange). We consider the range of amplitude ratios $A_{R,220} \in [0.001, 1]$. Right: The same quantity shown in a contour plot on the $(A_{R,220}, \delta\phi)$ plane for fixed $\chi_f = 0.67$.

## IV. FORECASTS WITH FUTURE OBSERVATIONS

In Fig. 2, we forecast the constraining power of this test by computing the minimum ringdown SNR, $\rho_{\rm det}$, for detectability of an extra scalar mode in the GR0+S model, for different amplitude ratios, remnant spins, and phase differences $\delta\phi = \phi_{220} - \hat{\phi}_{220}^{s=0}$. (Hereafter we focus on the scalar case, since the vector case gives qualitatively similar results, see Appendix VII C.) $\rho_{\rm det}$ is defined following Refs. [88, 95, 96], namely as the SNR such that $\sigma_{A_R} = A_{R,220}$, where the statistical error $\sigma_{A_R}$ has been computed with a fully numerical Fisher information matrix , also assuming that the relevant sensitivity curves at the ringdown regime are approximately flat for the two distinct final masses considered [95, 96]. In general, beyond-GR theories can introduce up to six polarization modes, by adding extra antenna-pattern response terms to the signal $h(t)$ [97]. Instead, we focus here on the corrections to the standard $h_\times$, $h_+$ GR polarizations, which are already detectable with current GW networks. Note that, in contrast to overtones [75, 88], *resolving* the frequency of the scalar mode is relatively easy since it is significantly different from the gravitational one. For this reason, $\rho_{\rm det}$ is always larger than the SNR threshold required to resolve the extra scalar mode. The requirements for the detectability and resolvability of a mode are outlined in the Appendix VII D.

We perform two variants of this analysis, either averaging over or fixing the sky-location of the signal, with results shown in the left and right panel of Fig. 2, respectively. Notice that, since the SNR is fully correlated

with the sky location, both approaches should yield consistent results (see Appendix VII D). Therefore, the right panel simply represents a phase-expanded version of the left-hand side plot.

The left panel of Fig. 2 shows that $\rho_{\rm det}$ decreases monotonically as $\chi_f$ increases – so highly spinning remnants favor this test – and that the phase difference $\delta\phi$ has a negligible impact for small amplitude ratios, while it can significantly affect the SNR at large amplitudes, especially for slowly-spinning remnants. The dependence on $\delta\phi$ can be better appreciated from the right panel of Fig. 2, showing a contour plot of the two-dimensional function $\rho_{\rm det}(A_{R,220}, \delta\phi)$ for $\chi_f = 0.67$. In the region of small amplitude ratios, the minimum SNR has a very simple scaling

$$\rho_{\rm det} \approx 4.0 \frac{g(\chi_f, \delta\phi, A_{R,220})}{A_{R,220}} . \qquad (7)$$

where $g \approx 1 + 0.25 A_{R,220} - 0.40\chi_f + 0.08\cos(\delta\phi + 1.16)$, and for $A_{R,220} \leq 0.12$ and $\chi_f \leq 0.9$ the fit is accurate within 10%.

In the left panel of Fig. 2 we also provide reference values of the ringdown SNR for current and future detectors. In particular, third-generation ground-based detectors [98] such as Cosmic Explorer [99–101] and the Einstein Telescope [102–104] are expected to observe a few events per year with ringdown SNR greater than 100 [96]. Likewise, GW space interferometers such as LISA [105] are expected to detect up to a dozen of massive BH mergers with ringdown SNR greater than 1000, depending on the massive BH population [95, 106]. Our results show that a ringdown SNR of 150 (resp., 1000) would yield a

constraint $A_{R,220} \lesssim 0.02$ (resp., 0.003) for $\chi_f \approx 0.9$, with only mild dependence on $\delta\phi$. Very similar results apply to the detectability of an extra vector mode.

This plot also confirms that the constraining power of the test is very limited when the ringdown SNR is around 10, which is a rough and even optimistic estimate for the previously analyzed events GW150914, GW190521, and GW200129.

While these results were obtained using a Fisher-matrix approximation to explore the entire parameter space, we have also compared individual points with synthetic injections at zero noise using the same Bayesian analysis discussed above for real-data events. At SNRs of $1.25\rho_{\text{det}}$ as given in Fig. 2, we find posteriors of $A_R$ peaking away from the lower bound and $\sigma_{A_R} \lesssim A_{R,220}$, already in agreement with the expectation from the Fisher analysis for the high-SNR limit.

## V. DISCUSSION

Any theory predicting extra ringdown modes should presumably also predict deviations from the standard gravitational Kerr QNMs, in which case one could argue that GW detectors are more sensitive to phase differences (and hence to QNM shifts) rather than amplitude differences, so that our test could have less constraining power than ordinary ringdown tests. However, there are known examples of theories predicting zero or negligible QNM shifts but extra modes, in which case our test can be superior to ordinary BH spectroscopy. An example is dynamical Chern-Simons gravity, wherein for a Schwarzschild BH the polar GR QNMs are unchanged, but the ordinary ringdown contains also extra scalar modes [32]. In the Kerr case also all the GR QNMs are modified, but the deviations are suppressed by powers of the BH spin [46], so the convenience of our method will likely depend on the remnant's spin.

The parametrization introduced in this work can also apply to cases in which the remnant is a Kerr BH or in the presence of extra fields but still within the realm of GR, provided the deviations from the Kerr case are sufficiently small. For example, it applies to Kerr–Newman BHs with small charge $Q_f$. In the perturbative limit, $Q_f/M_f \ll 1$, the parameter $Q_f/M_f$ effectively plays the role of a coupling constant in an effective field theory, making Eq. (5) applicable in this scenario, upon replacing the scalar sector with that of a test spin-1 field.

We focused on $\ell = m = 2$ but clearly, depending on the source, one should also add $\ell = 3$ modes, or overtones. In this context, it is noteworthy that the ringdown analysis for the latest GW event, GW250114 [107, 108], was mainly performed precisely with $\ell = m = 2$ modes only, using the fundamental mode and the first overtone. We point out that, for events such as GW250114 (but also for GW150914) a ringdown tests of GR *can* be based on $\ell = m = 2$ only, but it would be inconsistent to include QNM shifts alone, without adding extra modes.

Also, note that nearly equal-mass binaries are expected to dominate ringdown tests in both current and future GW detectors. For instance, using the fits of Ref. [88], binaries that are nearly face-on ($\iota < 30°$) with mass ratios $q \lesssim 1.2$ and effective spins $\chi_{\text{eff}} \sim 0.7$ exhibit amplitude ratios of approximately $(A_{33}Y_{33}/A_{22}Y_{22}) \sim 0.03$ and $(A_{22,\text{s}}Y_{22}/A_{22}Y_{22}) = A_{22,\text{s}}/A_{22}$. Therefore, values of $A_{22,\text{s}}/A_{22}$ as low as $\sim 0.03$ would still produce a secondary component comparable in strength to the $(3,3)$ mode. A similar argument applies to the $(2,1)$ mode, since in the same parameter regime one finds $(A_{21}Y_{21}/A_{22}Y_{22}) \sim 0.03$. Regarding overtones, given their short damping times, their relevance can be reduced by starting the ringdown analysis later (when scalar modes are still relevant). Clearly, this would reduce the overall SNR of the effective ringdown used for the test but would also avoid the data-analysis subtleties associated with overtones [66, 77].

As future extensions, it would be interesting to consider a specific theory and compare the constraints on the coupling constant(s) placed by our test with those of ordinary BH spectroscopy. This would require estimates of both ordinary QNM shifts and excitation amplitudes of extra modes in a given theory, both of which have recently become available for some theories [48–50, 109–116]. As an order-of-magnitude estimate, for a theory which adds quadratic curvature corrections to the standard Einstein-Hilbert action, the coupling constant $\alpha$ has the dimension of a length squared [6] In such a case one would expect $A_{R,220} = \mathcal{O}(1)\alpha^2/M_f^4$. Our results suggest that future detectors will be able to probe $A_{R,220} = \mathcal{O}(10^{-3})$, and hence a coupling constant as small as $\alpha^2/M_f^4 \sim \mathcal{O}(10^{-3})$.

A related extension is to include higher harmonics in the test. Besides adding complexity to the mode, this would also require including the inclination and the sky localization as extra waveform parameters. Work in this direction is underway.

Finally, another possible avenue of exploration is to use the recently introduced QNM filtering technique [36, 37, 80] to search for extra (scalar, vector, etc) modes.

## VI. ACKNOWLEDGEMENTS

We thank Collin Capano and Alexander Nitz for clarifications on PyCBC inference, and Gregorio Carullo, Cecilia Chirenti, Gabriele Franciolini, Nicolas Yunes, and the participants of the workshop Ringdown Inside and Out for insightful comments and discussions. This work is partially supported by the MUR PRIN Grant 2020KR4KN2 "String Theory as a bridge between Gauge Theories and Quantum Gravity", by the FARE programme (GW-NEXT, CUP: B84I20000100001), and by the INFN TEONGRAV initiative. X. Jimenez is supported by the Spanish Ministerio de Ciencia, Innovación y Universidades (Beatriz Galindo, BG22-00034) and cofinanced by UIB; the Spanish Agencia Estatal

de Investigación Grants No. PID2022-138626NB-I00, No. RED2022- 134204-E, and No. RED2022-134411-T, funded by MCIN/AEI/10.13039/501100011033/FEDER, UE; the MCIN with funding from the European Union NextGenerationEU/PRTR (No. PRTR-C17.I1); the Comunitat Autònoma de les Illes Balears through the Direcció General de Recerca, Innovació I Transformació Digital with funds from the Tourist Stay Tax Law (No. PDR2020/11 - ITS2017-006), and the Conselleria d'Economia, Hisenda i Innovació Grant No. SINCO2022/6719. Some numerical computations have been performed at the Vera cluster supported by the Italian Ministry of Research and by Sapienza University of Rome, and with the University of Birmingham's Blue-BEAR HPC service.

## VII. APPENDIX

### A. Explicit example: Dynamical Chern-Simons gravity

We provide here a specific example of a nonminimal coupling giving rise to extra scalar modes in the gravitational sector. We consider a theory with quadratic curvature corrections, dynamical Chern-Simons gravity, described by the action [26]

$$
S = \frac{1}{16\pi} \int d^4x \sqrt{-g} R - \frac{1}{2} \int d^4x \sqrt{-g} g^{ab} \nabla_a \vartheta \nabla_b \vartheta
$$
$$
+ \frac{\alpha}{4} \int d^4x \sqrt{-g} \vartheta \, {}^*RR \,. \tag{8}
$$

where $\vartheta$ is the scalar field, ${}^*RR = \frac{1}{2} R_{abcd} \epsilon^{baef} R^{cd}{}_{ef}$ is an odd-parity quadratic-curvature invariant, and $\alpha$ is the coupling constant, with dimensions of a squared mass (henceforth we adopt $G = c = 1$ units).

As in the GR case, the only stationary, spherically-symmetric solution is the Schwarzschild metric. For simplicity we consider perturbations of this solution, neglecting the spin of the background. Axial perturbations of the metric are coupled to those of the scalar field. Upon a spherical harmonic decomposition and in the frequency domain, they reduce to the following set of coupled ordinary differential equations [32]

$$
\begin{cases}
\frac{d^2}{dr_\star^2}\Psi + \left\{ \omega^2 - f\left[ \frac{l(l+1)}{r^2} - \frac{6M}{r^3} \right] \right\} \Psi = \frac{96\pi M f}{r^5}\alpha\Theta, \\
\frac{d^2}{dr_\star^2}\Theta + \left\{ \omega^2 - f\left[ \frac{l(l+1)}{r^2}\left( 1 + \frac{576\pi M^2\alpha^2}{r^6} \right) + \frac{2M}{r^3} \right] \right\} \Theta = f\frac{(l+2)!}{(l-2)!}\frac{6M\alpha}{r^5}\Psi
\end{cases} \tag{9}
$$

where $f(r) = 1 - 2M/r$ and $r_\star \equiv r + 2M \ln(r/2M - 1)$ is the standard Schwarzschild tortoise coordinate. The variables $\Psi$ and $\Theta$ reduce to the standard metric and scalar master functions, respectively, in the decoupling limit, $\alpha \to 0$. Indeed, when $\alpha = 0$ the two equations decouple and reduce to the standard Regge-Wheeler equation and scalar-perturbation equation of a Schwarzschild BH, respectively. However, when $\alpha \neq 0$ the two perturbations are coupled to each other and the scalar effective potential acquires some corrections.

The coupling $\alpha$ gives rise to two features [32, 34]:

1. The above system of equations contains both gravity-led and scalar-led modes, both displaying $\mathcal{O}(\alpha^2)$ corrections with respect to GR:

$$
\omega = \omega^{\text{GR, grav}}\left( 1 + \gamma^{\text{grav}}\frac{\alpha^2}{M^4} \right), \tag{10}
$$

$$
\hat{\omega} = \omega^{\text{GR, scal}}\left( 1 + \gamma^{\text{scal}}\frac{\alpha^2}{M^4} \right), \tag{11}
$$

where $\omega = 2\pi f - i/\tau$ are the complex QNM frequencies, $\omega^{\text{GR, grav}}$ are the standard Schwarzschild QNMs in GR, $\omega^{\text{GR, scal}}$ are the test-scalar QNMs of Schwarzschild, whereas $\gamma^{\text{grav}}$ and $\gamma^{\text{scal}}$ are dimensionless order-unity constants, the value of which depends on the overtone number $n$.

2. The above system of equations is akin to a coupled harmonic oscillator, so a scalar perturbation would source scalar modes in the gravitational sector, and vice versa. In particular, in this theory the scalar field is at least linear in $\alpha$ [26], in which case the coupled perturbation equations imply that, whether or not scalar perturbations are present in the merger conditions (for example if the progenitors are endowed with a scalar field [117]), the amplitude of the scalar mode in the gravitational sector would be $\mathcal{O}(\alpha^2)$. In the notation of the main text, this would imply that, at the leading order, $A_{R,220} = \gamma \cdot \alpha^2/M_f^4$, where $\gamma$ is a source-dependent excitation factor [32, 46, 50, 118].

These arguments show that the *gravitational* ringdown in this theory can be schematically modelled as

$$
h(t) = \sum_j A_j e^{i(\omega_j t + \phi_j)} + \sum_j \hat{A}_j e^{i(\hat{\omega}_j t + \hat{\phi}_j)}, \tag{12}
$$

where the sum runs over the overtones. The scalar modes $\hat{\omega}_j$ contains $\mathcal{O}(\alpha^2)$ corrections, but $\hat{A}_j$ is at least $\mathcal{O}(\alpha)$ or higher. Therefore, to leading order in $\alpha$ we can approximate $\hat{\omega}_j \approx \omega_j^{\text{GR, scal}}$ in the second term of the above equation, which is the crucial simplification of our test.

This is consistent with the finding of Ref. [32], where the gravitational ringdown in the $\alpha \ll M^2$ limit contains both the *unperturbed* gravitational and scalar modes.

While we explicitly showed this for a specific theory, it is in fact a very general properties of extended theories of gravity with nonminimal couplings (see, e.g., [57] for another example).

### B. The effect of precession on GW190521 and GW200129

There are well-founded arguments suggesting that GW190521 and GW200129 could originate from precessing binaries [82, 84]. As opposed to aligned-spin systems, precession breaks the symmetry between $m \leftrightarrow -m$ angular modes, resulting in $h_{lm} \neq (-1)^l h_{l-m}^*$. In the main text, we searched for a scalar mode using a $(lm) = (2, \pm 2)$ spin-aligned (non-precessing) waveform. Here, we extend the parameter estimation on GW190521 and GW200129 including precession, and varying $t_{\text{offset}} = [-2, 0, 2]$ms. This is achieved by allowing the amplitude $A_{22n} \neq A_{2-2n}$ and the phase $\phi_{22n} \neq -\phi_{2-2n}$, for both the gravitational and scalar mode. In Fig. 3, we show the corner plots for GW200129 with $t_{\text{offset}} = 0$, for both the GR1 and GR0+S cases. Notice that, since only $(lm) = (2, \pm 2)$ modes are used, accounting for precession in this case does not significantly affect the values of the mass and the spin of the remnant compared to the aligned-spin case. However, including precession affects the amplitude of the fundamental mode, by increasing its uncertainty. We find qualitatively similar results for the other offset times and for GW190521.

### C. Supplemental results

In Fig. 4 we show the frequencies and damping times of some representative QNMs, including scalar, vector, and gravitational modes as functions of the remnant's spin. Note that the frequency of the (220) scalar or vector mode is always well separated from that of the fundamental gravitational mode, while its damping time is comparable. These are advantages with respect to overtones, which are instead harder to resolve and decay more rapidly.

Figure 5 presents an example of posterior distributions of some waveform parameters obtained from our Bayesian analysis on real data, namely the case of GW150914 analyzed with various models. Although, as discussed in the main text, there is no statistical evidence for extra scalar modes in the data, their inclusion in the parameter estimation affects the posterior of $\chi_f$ and $A_{220}$.

Further, in Fig. 6, we compare some representative posterior distributions obtained from the Bayesian inference on real data with forecasts using injections at higher SNR (equal to 100). We consider a ringdown model with

an extra scalar mode (GR0+S, left panel) and with an extra vector mode (GR0+V, right panel). For the injection simulations, we inject the values corresponding to the maximum likelihood values of the real data. We notice that, at higher SNRs, the precision of the parameters' distribution improves, and the signal is well reconstructed. This can be easily observed with the distributions of $\chi_f$, which in the case of the real data is spread, due to the effect of the gated Gaussian noise. Also in this case, we note that the results for the scalar or vector case are very similar.

Finally, we have searched for multi-modality in the amplitude of the scalar mode. This is supported by the fact that, for some different choices of mass and spin, the fundamental scalar/vector mode can mimic the fundamental gravitational QNM. In general, this effect is not evident due to the limited range of the prior on the amplitude ratio $\hat{A}_{220} \in [0, 10]$. We have thus performed a new run where we inject GR0 with GW150914-like parameters (but in our case, we have $SNR = 83$), and recover with GR0+S, but with a broader prior on the amplitudes ($A_{220}, \hat{A}_{R,220} \in [0, 100]$), indeed finding bimodality. Including a secondary mode in the ringdown analysis, as we did in certain cases, clearly breaks this degeneracy.

### D. The detectability and resolvability criteria

A BH no-hair test is designed to check the consistency of the mass and spin as observed from two different ringdown modes: the fundamental mode ($l = 2, m = 2, n = 0$), and a next-to-leading order mode with either $|l| = |m| \neq 2$ or $n \neq 0$. Ideally, one would have to infer to significant accuracy the tetrad of parameters $\{\omega_{220}, \tau_{220}, \omega_{lmn}, \tau_{lmn}\}$. In practice, one just needs to build a triad $\{\omega_{220}, \tau_{220}, q_{lmn}\}$ with $q_{lmn} = \omega_{lmn}$ or $q_{lmn} = \tau_{lmn}$, and check the consistency on the inferred mass and the spin from the following tuples $\{(\omega_{220}, \tau_{220}), (\omega_{220}, q_{lmn}), (\tau_{220}, q_{220})\}$ [4]. Whether we can reliably distinguish a secondary mode depends critically on its SNR. The threshold SNR can be defined in different ways as:

1. $\rho_{\text{det}}$: The ringdown SNR at which the amplitude of the mode is large enough compared to its statistical uncertainty $\sigma_{A_R}/A_{lmn}^R \sim 1$.

2. $\rho_{\text{res}}$: The ringdown SNR at which the spectrum $q_{lmn}$ subtracted to the spectrum $q_{220}$ of fundamental mode is comparable to its statistical uncertainty $\sigma_{q_{lmn}}/|q_{220} - q_{lmn}| \sim 1$,

where $\sigma_{A_R}$ and $\sigma_{q_{lmn}}$ are the $1 - \sigma$ statistical errors on $A_R$ and $\sigma_{q_{lmn}}$ respectively [119]. A mode is measured if conditions 1) and 2) are satisfied, which will occur for $\rho \gtrsim \max\{\rho_{\text{det}}, \rho_{\text{res}}\}$ [3, 119]. Using the Fisher Matrix approach, the set $\{\rho_{\text{det}}, \rho_{\text{res}}\}$ can be estimated analytically, provided that the sensitivity curve in the ringdown regime is sufficiently flat for a given total mass. The

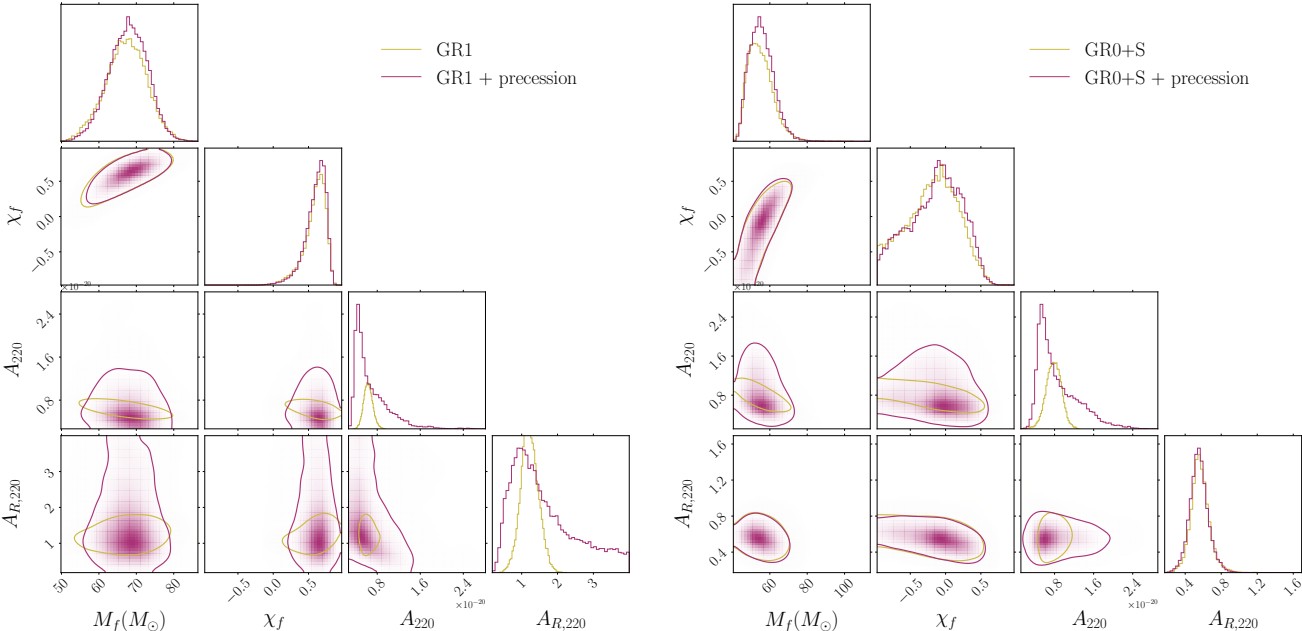

FIG. 3. Posterior distributions for the event GW200129 with $t_{\text{offset}} = 0$ms for the models GR1 (left) and GR0+S (right). In both cases, the case without (resp., with) precession is indicated in yellow (resp., purple).

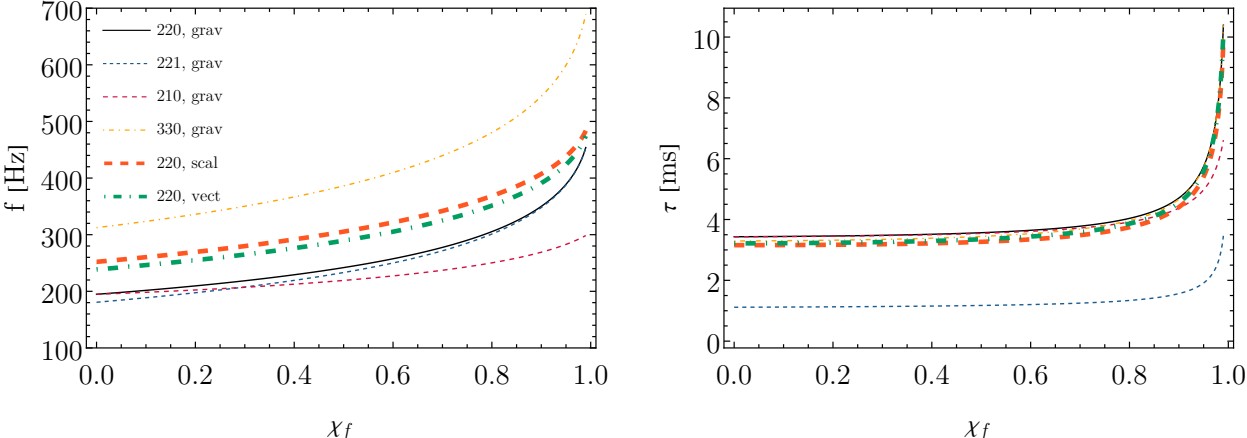

FIG. 4. Frequencies (left) and damping times (right) for a few gravitational modes and for (220) scalar and vector modes of Kerr as a function of the spin. Results are normalized for a remnant mass compatible with GW150914, $M_f = 62\,M_\odot$.

statistical error $\sigma_{q_{lmn}}$ and the SNR of a signal $\rho$ are computed as,

$$\sigma_{ij}^2 \propto \left( \int \frac{\frac{dh}{d\lambda_i}\left(\frac{dh}{d\lambda_j}\right)^*}{S_n(f)} df \right)^{-1} \quad , \quad \rho^2 \propto \int \frac{|h|^2}{S_n(f)} df \,, \tag{13}$$

where $\lambda_{i,j}$ are each of the ringdown intrinsic parameters, $^*$ denotes the complex conjugate and $^{-1}$ the matrix inverse. Note that the dependence of the waveform on the of extrinsic parameters $f(\Theta)$ will be the same for both equations since one can replace $h \to h \cdot f(\Theta)$. The GW ringdown spans over a relatively short frequency range. For the scalar mode case, and for $M_f = 62$, notice that the whole ringdown signal is approximately all contained within $[200, 300]$ Hz at $a_f \sim 0.7$ (see Fig. 4). Therefore, one can assume an effective $S_n(f) \sim S_n(f_0)$ where $f_0$ is an appropriate frequency within the range described above. The last approximation allows us to define $f(\lambda_{ij}) \equiv \rho_{(\text{det,res})} \cdot \sigma_{ij}$, which only depends on the intrinsic parameters $\lambda_{ij}$, and allows us to also describe both the $\rho_{\text{det}}$ and $\rho_{\text{res}}$ independent from the intrinsic parameters as

$$\rho_{\text{det}} = \frac{f(\lambda_{A_{lmn}^R})}{A_{lmn}^R} \quad , \quad \rho_{\text{res}} = \frac{f(\lambda_{q_{lmn}})}{|q_{220} - q_{lmn}|} \,. \tag{14}$$

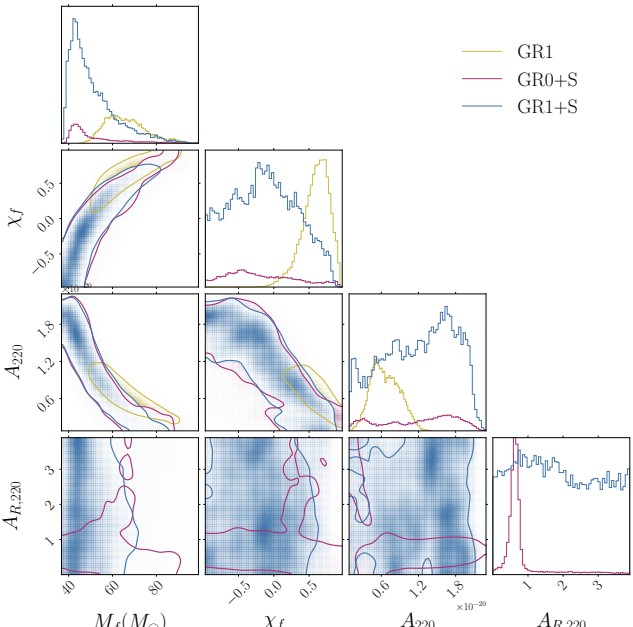

FIG. 5. Posterior distributions of some ringdown parameters for GW150914 in our analysis: $M_f$ and $\chi_f$ are the remnant's mass and spin, $A_{220}$ is the amplitude of the fundamental gravitational mode, and $A_{220}^R$ is the amplitude ratio of the (220) scalar mode relative to the fundamental gravitational one. We consider $t_{\text{offset}} = 2\,\text{ms}$ and the GR1 (magenta), GR0+S (yellow), and GR1+S (cyan) models. The contours state the 90% credible levels.

For scalar and vector modes, we obtain that $\rho_{\det} \gg \rho_{\text{res}}$ for $A_{lmn}^R \lesssim 1$. Therefore, $\rho > \rho_{\det}$ is our determining criterion for observing them. The above equation also shows clearly the trend $\rho_{\det} \sim (A^R)^{-1}$ displayed in Fig. 2 of the main text. Similarly, a $M_f \sim 10^7 M_\odot$ binary will cover a short frequency range at the mHz, in which current estimates of the LISA sensitivity curve are rather flat [105]. Finally, notice that both $\rho_{\det}$ and $\rho_{\text{res}}$ scale with $f \propto \rho \cdot \sigma_{ij}$. For a binary system considering only the $(\ell, m) = (2, 2)$ mode, non-precessing, and analyzed in the frequency domain, the GW strain can be expressed as $\tilde{h} = H_{22}(\vec{\Theta})h_+(\vec{\lambda})e^{i\phi_{22}(\vec{\lambda})}$ where $H_{22}$ is an amplitude factor that encodes the sky position, polarization angle, and distance, $h_+(\vec{\lambda})$ is the plus polarization component, $\phi_{22}$ is the phase, and $\vec{\lambda}$ represents the intrinsic parameters of the system [120]. Therefore, it follows that the product $f \propto \rho \cdot \sigma_{ij}$ is independent of the system's sky position, implying that both $\rho_{\det}$ and $\rho_{\text{res}}$ are also independent of $\vec{\Theta}$.

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

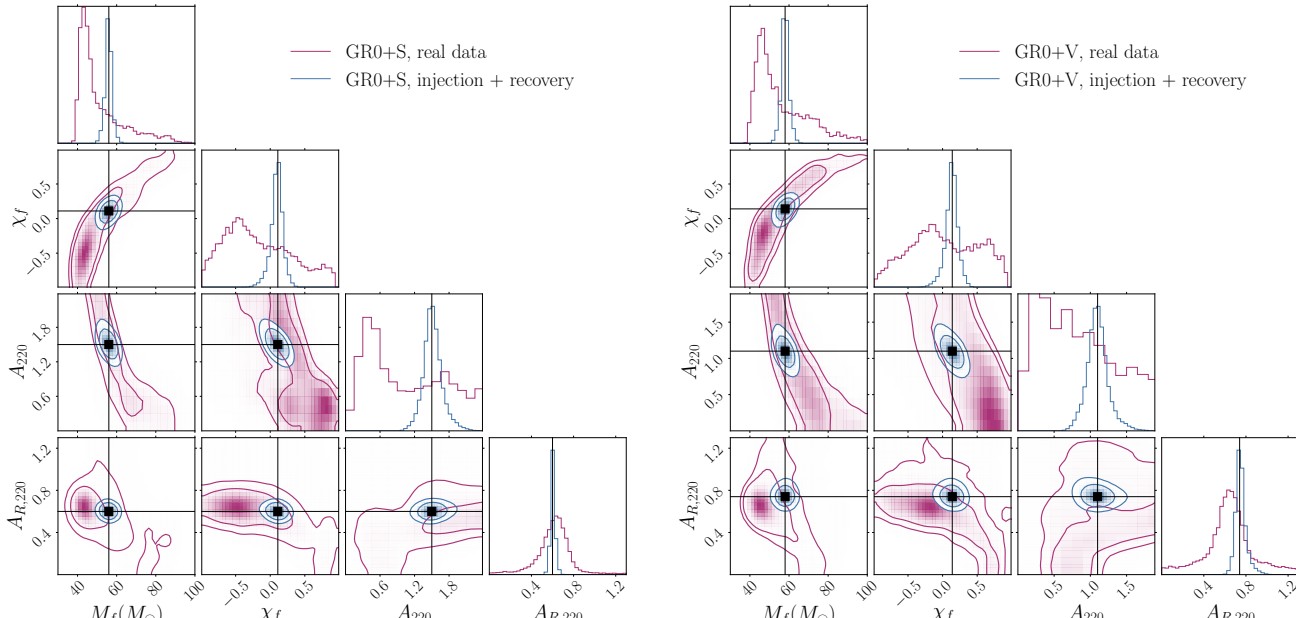

FIG. 6. Posterior distributions for real data (magenta) vs injection+recovery (blue) parameter estimation simulations with model `GR0+S` (left) and `GR0+V` (right). The levels of the 2D distributions indicate the 68% and the 90% credible intervals, while the black lines correspond to the injected values. Note that, in this case, we indicate the fundamental amplitude $A_{220}$ rescaled by $d_L/M_f^{(m)}$, where $M_f^{(m)}$ is the final mass of the remnant expressed in meters.

JHEP **09**, 122 (2017), arXiv:1704.01590 [gr-qc].

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

[80] S. Ma, L. Sun, and Y. Chen, Phys. Rev. D 107, 084010 (2023), arXiv:2301.06639 [gr-qc].

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
