# Peer review of "Theory-agnostic searches for non-gravitational modes in black hole ringdown"

_SciPost Physics_

## Round 1 · Referee Report · Anonymous (Referee 1) · 2025-8-26

Report

The paper studies the possibility of testing GR by detecting additional modes in the quasinormal mode ringing following binary black hole merger. The study is made theory agnostic, namely certain reasonable assumptions are made for the QNM spectrum without specifying the underlying theory of gravity. This is different from the typical approach where deviations from GR are probed via deformation/small deviations from the Kerr QNM spectrum. The paper is in general interesting but I have several important concerns:
1. The amplitudes of such scalar mode induced gravitational radiation should be very small. The authors should take good care to estimate whether it is even theoretically observable, taking into account the present observational constraints on the coupling parameters (in certain well-studied examples of relevant modified gravity theories). It is important to know whether the amplitudes would be larger compared to other effects, such as nonlinearities and overtones, for example. In general, the part with excitation amplitudes and their relation to the coupling parameters should be much better explained.
2. One of the very weak points in the paper up to my opinion is neglecting higher l,m,n modes in the spectrum. It is a well-known fact that for parameter estimation (especially for unequal mass binaries) and QNM ringdown fitting, it is not enough to use only the 2,2 modes. Therefore, I am not convinced in the validity of such a restrictive approach.

Recommendation

Ask for major revision

  • validity: high
  • significance: good
  • originality: good
  • clarity: good
  • formatting: excellent
  • grammar: perfect

Author:  Francesco Crescimbeni  on 2025-10-20  [id 5946]

(in reply to Report 1 on 2025-08-26)

The referee writes: "The amplitudes of such scalar mode induced gravitational radiation should be very small. The authors should take good care to estimate whether it is even theoretically observable, taking into account the present observational constraints on the coupling parameters (in certain well-studied examples of relevant modified gravity theories). It is important to know whether the amplitudes would be larger compared to other effects, such as nonlinearities and overtones, for example. In general, the part with excitation amplitudes and their relation to the coupling parameters should be much better explained."

Our response: As shown by previous work and summarized in Appendix A.1, in any effective field theory (EFT) extensions of GR the amplitude of these extra modes enter at the same order as the standard corrections to the gravitational QNMs. For example, for quadratic theories like dCS/EsGB gravity, the amplitude of the scalar is γ · α^2/M^4 , where γ is an excitation factor depending on the initial data, and α is the coupling constant. Note that the QNM corrections in these theories are also proportional to α^2/M^4 so the two effects are of the same order. Highlighting this fact and quantify the errors in neglecting the extra modes was one of our main goals. Indeed, an independent paper that appeared after our original submission [4] has shown that the two effects (extra modes and QNM corrections) are ubiquitous beyond GR and estimated that they can be equally relevant to model the GW phase. This conclusion is also confirmed and quantified by a very recent follow-up paper by some of us [2]. In general, the amplitude of the scalar modes is theory-dependent and also depends on the initial conditions giving rise to the merger. For this reason, it is not easy to see if the scalar mode is dominant with respect to, e.g., nonlinearities, although this would be an interesting follow-up work. Here, we consider the amplitude to be a free parameter to remain agnostic. Regarding overtones, given their short damping time, their relevance can be reduced by starting the ringdown analysis later (where scalar modes are instead still relevant). Clearly this would reduce the overall SNR of the effective ringdown used for the test, but would also avoid the data-analysis subtleties related to overtones.

The referee wirtes: "One of the very weak points in the paper up to my opinion is neglecting higher l,m,n modes in the spectrum. It is a well-known fact that for parameter estimation (especially for unequal mass binaries) and QNM ringdown fitting, it is not enough to use only the 2,2 modes. Therefore, I am not convinced of the validity of such a restrictive approach."

Our response: We understand the Referee’s comment, but would like to stress that our goal was not to include all possible effects; rather, we want to highlight that, when performing a ringdown tests of gravity, it is intrinsically unmotivated to include QNM shifts and not extra modes. Clearly, depending on the source, one should also add ℓ = 3 modes, or overtones. In this context, it is noteworthy that the ringdown analysis for the latest LVK event, GW250114, was mainly performed precisely with ℓ = m = 2 modes only, using the fundamental mode and the first overtone. Our point is that, for events such as GW250114 (but also for GW150914) a ringdown tests of GR can be based on ℓ = m = 2 only, but it would be inconsistent to include QNM shifts (as done, e.g., in [1]) alone, without adding extra modes. We agree that, especially for asymmetric binaries, other harmonics are important, and indeed including them was listed as an important extension. Since the time of original submission, we have been working on this problem and the related analysis is included in a recent follow-up [2], which confirms our original conclusions. Note that nearly equal-mass binaries are expected to dominate ringdown tests in both current and future gravitational-wave detectors. For instance, using the fits of Ref. [3], binaries that are nearly face-on (ι < 30◦) with mass ratios q ≲ 1.2 and effective spins χeff ∼ 0.7 exhibit amplitude ratios of approximately (A_33 Y_33/A_22 Y_22) ∼ 0.03 and (A_22,s Y_22/A_22Y_22) = A_22,s/A_22. Therefore, values of A_22,s/A_22 as low as ∼ 0.03 would still produce a secondary component comparable in strength to the (3, 3) mode. A similar argument applies to the (2, 1) mode, since in the same parameter regime one finds (A_21Y_21/A_22Y_22) ∼ 0.03.

References [1] Adrian Ka-Wai Chung and Nicol´as Yunes. Probing quadratic gravity with black-hole ringdown gravitational waves measured by LIGO-Virgo-KAGRA detectors. 6 2025. [2] Francesco Crescimbeni, Xisco Jimenez-Forteza, and Paolo Pani. Black Hole Ringdown Amplitudescopy. 10 2025. [3] Xisco Jim´enez Forteza, Swetha Bhagwat, Paolo Pani, and Valeria Ferrari. Spectroscopy of binary black hole ringdown using overtones and angular modes. Phys. Rev. D, 102(4):044053, 2020. [4] Jacopo Lestingi, Giovanni D’Addario, and Thomas P. Sotiriou. Frequency contamination from new fundamental fields in black hole ringdowns. Phys. Rev. D, 112(6):064070, 2025.

---

## Round 1 · Referee Report · Anonymous (Referee 2) · 2025-9-23

Strengths

1-Interesting idea

2-Concisely communicated

Weaknesses

1-Could be more systematic
(for instance in choice of gravitational wave events and delineation of model applicability)

2-Could work more towards definite conclusions
(for instance, instead of "... in which case one could argue that GW detectors are more sensitive to phase differences (and hence to QNM shifts) rather than amplitude differences, so that our test could have less constraining power than ordinary ringdown tests." study explicitly an example in which this can be shown to be the case. Instead of "In the Kerr case also all the GR QNMs are modified, but the deviations are suppressed by powers of the BH spin, so the convenience of our method will likely depend on the remnant's spin." compare as a function of spin and determine the range of spins for which it is. Compare also to other (inspiral etc.) tests.)

3-key analyses contributing to conclusions about the validity and relevance of the work are relegated to future work (for instance, inclusion of other modes and verifying specific model in which the test could have more power than ordinary ringdown test)

Report

Please see attachment.

Attachment

Recommendation

Ask for minor revision

  • validity: good
  • significance: good
  • originality: high
  • clarity: high
  • formatting: excellent
  • grammar: excellent

Author:  Francesco Crescimbeni  on 2025-10-20  [id 5947]

(in reply to Report 2 on 2025-09-23)

The referee writes: "While the test proposed by the authors is not entirely model-specific, it is also not entirely theory- agnostic or “generic”. Therefore, I would first ask if the authors agree with my reading summarized above that the tests are appropriate for “theories of gravity which are perturbatively close to General Relativity with additional minimally coupled scalar or vector fields for final compact objects perturbatively close to Kerr black holes”. Second, I would then propose for the discussion to try to delineate more clearly when the authors believe these tests would be appropriate. See also the following point for classes of examples which seem to not be mentioned in the manuscript."

Our response: Regarding the first part of the Referee’s comment, we do agree with the assessment that our test assumes “theories of gravity which are perturbatively close to General Relativity”. Note that the rest (i.e. additional fields and final objects perturbatively close to Kerr) are not assumptions but rather consequences of the quoted statement. Indeed, by Lovelock’s theorem, almost any theory of gravity can be formulated as GR plus extra fields. For example, f (R) gravity does not manifestly contain extra fields, but can be reformulated as a scalar-tensor theory. Likewise, in massive gravity and quadratic gravity one finds extra scalar, vector, tensor degrees of freedom. This is why extra fields (minimally or nonminimally coupled to gravity) are essentially unavoidable in beyond GR theories. The assumption of “being perturbatively close to GR” is motivated both by the well-established EFT approach to gravity (wherein GR is the leading term of an EFT expansion) and by observational constraints (showing that LVK events cannot be too far from GR predictions). The EFT is the standard framework to treat corrections rigorously so we do not consider this as a strong limitation. Note also that, within this assumption, the remnant must be perturbatively close to Kerr. This is indeed the case in those theories that can be studied in details: Cherns-Simons gravity, Einstein-scalar-Gauss-Bonnet gravity, EFT theories with cubic and quartic curvature terms, etc. We also note that, in principle, extra modes will exist also beyond the EFT regime. However, our test relies on the fact that, within a perturbative regime, the extra modes can be consistently approximated as those of test fields, making the analysis as much model-agnostic as possible. Beyond the EFT, this simplifying feature won’t be justified. Regarding the second part of the Referee’s comment, we have extended the discussion on these points in the revised manuscript. We hope these are now properly clarified.

The referee writes: "After motivating ringdown tests in the introduction as (page 1, left-column, last paragraph): “This provides opportunities for conducting multiple null-hypothesis tests of gravity [15,16] and investigating the nature of the remnant [17-19]”, the emphasis of the manuscript seems decidedly on tests of gravity with additional degrees of freedom non-minimally coupled to gravity as opposed to the nature of the remnant. On the other hand, it seems to me that various theories with additional degrees of freedom minimally coupled to gravity may have the same signature depending on the nature of the remnant. Consider General Relativity coupled to an electromagnetic field. If the remnant has a charge which is not too big compared to the mass, there are clear gravitationally-led modes perturbed from Kerr as in the first line of (3) of the manuscript and electromagnetically-led modes behaving as the second line of (3), with approximately the free vector QNM frequencies. A similar setup giving rise to scalar-led mode contributions, which to leading order would have a gravitational piece with free scalar frequencies, could occur for minimally- coupled scalar fields with self-interactions for a remnant with scalar hair. Do the authors agree and have they considered this? Clearly, there would be other ways to look for such a final charged or otherwise hairy remnants but without being more specific about the model, it is not clear from the analysis in the manuscript that the same could not be said for many non-minimally coupled examples as well."

Our response: The Referee raised an interesting point: if the remnant is an object different from a (Kerr) BH, does our method apply? Similarly to what discussed above, also in that case we expect extra modes (e.g., if the remnant is a boson star there will be scalar-driven modes), but since in general the remnant will not be perturbatively close to Kerr, the extra modes cannot be taken to be those of a test field, so the actual implementation would be model dependent. Likewise, the test can be applied also to Kerr-Newman BHs, provided the charge Q is small enough. Indeed, taking Q/M ≪ 1, the parameter Q/M would play the role of the coupling constant in an EFT and, to leading Q^2/M ^2 order, both the amplitude of test vector modes will affect the GW signal and the QNM frequencies will get shifted. We added a comment on these interesting points.

The referee writes: "In Appendix D, it is not entirely clear to me what is meant by “indistinguishably” in “Our ability to indistinguishably observe a secondary mode ...”. From ρres, I would infer that it is meant to imply the secondary mode is distinguishable from the fundamental gravitational 220 mode, is that correct? REPLY: We apologize for the confusion, we have rephrased to “Whether we can reliably distinguish a secondary mode depends critically on its SNR.” We agree with the Referee’s statement. 4. A key point emphasized in the manuscript is how different the scalar or vector modes are in frequency from the gravitational, Kerr modes in order to be “distinguishable” (see previous point). Here, the focus seems to be on the comparison between the fundamental gravitational mode and other modes: “At variance with overtones, the frequency of the (220) scalar or vector mode is always well separated from that of the fundamental gravitational mode (and hence more easily resolvable from the latter)” (page 3, end of section II). First, this seems to ignore the degeneracy of scalar, vector, and gravitational (220) modes as χ_f → 1. Second, in the context of tests of black holes in General Relativity, I would be more concerned about misidentifying another gravitational, Kerr mode or quadratic QNM with a scalar or vector mode. Why are the authors not more concerned about such confusion? More generally, one may worry that the proposed test only becomes interesting at ringdown SNR where several other modes, currently neglected, should be included in the analysis."

Our response: The Referee raised two points, addressed below. 1. We agree with this point. However, in our case, the degeneracy is never reached, since the value of χf = 1 is always excluded in the posterior distributions. For the typical values of merger remnants detected so far, the different modes are still quite separated from each other. We have added a footnote on page 3 to clarify this point. 2. We also agree with this point. However, we emphasize that even at higher SNR—where the test is naturally more sensitive—there exists a subclass of systems for which these modes are expected to be weaker than a possible scalar- or vector-driven mode. For instance, face-on and nearly symmetric systems tend to suppress the amplitude of higher modes (see, e.g., Ref. [2]). Moreover, overtones with n > 0 decay much faster than the fundamental (n = 0) scalar and gravitational modes, so the late-time ringdown signal is expected to be dominated by the n = 0 contributions. Regarding quadratic modes, consider Ref. [4], in particular their Fig. 16. That figure shows the number of events expected to yield a confident detection of a quadratic mode after applying various selection cuts (e.g., on SNR, mass ratio, etc) for three astrophysically motivated models. As can be seen in the top row, the number of such events is very small, indicating that in this case non- linearities are unlikely to be observable.

The referee writes: "Currently, three events are considered for the real data. For each of the events, some motivation is given for why they are an interesting choice in the beginning of Section III but do the authors have a more systematic selection mechanism (say ringdown SNR)? Why not?"

Our response: The three events considered in our analysis are highly representative of the ringdown phase, as discussed in the main text (section III, first paragraph). GW150914 was (prior to the recent GW250114) the prototypical event for ringdown tests, and the one in which tentative evidence for overtones was reported; GW190521 was the first event showing tentative evidence of a secondary ℓ = 3 mode; and GW200129 is a loud event exhibiting some tension with GR in certain inspiral–merger–ringdown tests. While we believe these are among the most relevant events for ringdown analyses (excluding GW250114, which was announced after the completion of our work), we stress again that our study is a proof of principle, and we did not aim to be exhaustive in the event selection.

The referee writes: "On page 4, right-column (main text, below the figure), Jeffreys’ scale criterion is invoked with reference to his book “The Theory of Probability” from 1939. Some more recent references within the domain of gravitational wave tests of General Relativity such as [1, 2, 3] may be more appropriate to interpret the results in addition to the historical reference, whose thresholds could be considered arbitrary in the context of the manuscript."

Our response: We thank the Referee for these suggestions; the corresponding references have been added to the bibliography.

The referee writes: "It seems the amplitude-ratio AR,220 is currently defined in the caption to Table I. If so, I would also define it in the main text."

Our response: We thank the Referee for this suggestion. We have added the definition of the amplitude ratio in the text.

The referee writes: "On page 2, left-column, last paragraph it is written that: “Namely, we propose to look for extra modes in the ringdown signal, which are not related to deformations of the Kerr ones”. Perhaps it is not necessary to emphasize this, but they are not related in frequency directly to deformations of the gravitational Kerr ones."

Our response: We agree with this observation, and changed the text accordingly.

References [1] Francesco Crescimbeni, Xisco Jimenez-Forteza, and Paolo Pani. Black Hole Ringdown Ampli- tudescopy. 10 2025. [2] Xisco Jim´enez Forteza, Swetha Bhagwat, Sumit Kumar, and Paolo Pani. Novel Ringdown Amplitude- Phase Consistency Test. Phys. Rev. Lett., 130(2):021001, 2023. [3] Jacopo Lestingi, Giovanni D’Addario, and Thomas P. Sotiriou. Frequency contamination from new fundamental fields in black hole ringdowns. Phys. Rev. D, 112(6):064070, 2025. [4] Sophia Yi, Adrien Kuntz, Enrico Barausse, Emanuele Berti, Mark Ho-Yeuk Cheung, Konstantinos Kritos, and Andrea Maselli. Nonlinear quasinormal mode detectability with next-generation gravita- tional wave detectors. Phys. Rev. D, 109(12):124029, 2024.

---

## Round 2 · Author Response

We thank the Referee for the time dedicated to reviewing our work and for their constructive and
insightful comments. We address their points below; the corresponding revisions in the manuscript are
highlighted in red.
As a general remark, we note that since our original submission to the arXiv (more than one year
ago), our results have been confirmed and extended in a recent study by other authors [1], as well as in
a follow-up work by some of us [2], further highlighting the relevance of our original findings.
We believe that the main contribution of our work was to clarify that additional modes in the GW
signal are at least as significant as shifts in the QNM frequencies—an effect largely overlooked in most
previous analyses, including the LIGO–Virgo–KAGRA ringdown tests of gravity. As detailed below, we
do not claim that other effects (such as overtones, nonlinearities, or higher harmonics) are negligible;
rather, their relative importance depends on the specific modified-gravity theory and, crucially, on the
nature of the source.
Sincerely,
The Authors
[1] Jacopo Lestingi, Giovanni D’Addario, and Thomas P. Sotiriou. Frequency contamination from new
fundamental fields in black hole ringdowns. Phys. Rev. D, 112(6):064070, 2025.
[2] Francesco Crescimbeni, Xisco Jimenez-Forteza, and Paolo Pani. Black Hole Ringdown Amplitudescopy. 10 2025.

---

## Round 2 · List of Changes



---

## Round 3 · Referee Report · Anonymous (Referee 1) · 2025-12-15

Report

The authors extended their paper according to my suggestions and I think it can be accepted for publication in its present form.

Recommendation

Publish (easily meets expectations and criteria for this Journal; among top 50%)

---

## Editorial Decision

in_voting